

# Life-history traits and intra-cohort divergence of clearhead icefish (*Protosalanx chinensis*), indicating cannibalism increased fitness

Fujiang Tang[1], Wei Liu[1], Jilong Wang[1] and James Henne[2]

[1] Heilongjiang River Fisheries Research Institute of Chinese Academy of Fishery Sciences, Harbin, China
[2] United States Fish and Wildlife Service, Bears Bluff National Fish Hatchery, Wadmalaw Island, South Carolina

## ABSTRACT

Cannibalism is considered one of the causes of intra-cohort size divergence in fish and is usually believed to result in increased fitness in terms of survival and reproduction, but direct evidence of this is lacking. Population demographics of the clearhead icefish (*Protosalanx chinensis*) from Lake Xingkai (Khanka) were investigated for one year. Size-frequencies exhibited a bimodal distribution from July through January, where the population diverged into an upper and a lower modal group based on size. Stomach content analysis confirmed the occurrence of cannibalism, where fish belonging to the larger, upper modal group preyed upon those of the smaller, lower modal group. We found *P. chinensis* does not spawn until all of the oocytes have reached maturity and then a single spawning event occurs although the oocytes may develope asynchronously in the ovary. Upper modal group females matured slightly earlier than those of the lower modal group, and reproductive investment was considerably greater in the upper modal group than the lower modal group. The lower modal males made up the majority of the population after the spawning period, which meant they may have few opportunities to participate in reproduction. Therefore, piscivory and cannibalism of *P. chinensis* appears to have increased fitness of the fish belonging to the upper modal group and greatly reduced the fitness of those belonging to the lower modal group.

Corresponding author
Fujiang Tang, tangfujiang@hrfri.ac.cn, rivery2008@163.com

## INTRODUCTION

Many piscivorous fishes exhibit an ontogenetic dietary shift, transitioning from non-fish prey to piscivory over time. An early transition to piscivory is thought to result in increased growth and reproductive fitness, decreased mortality, and therefore enhanced lifetime fitness (*Werner & Gilliam, 1984*; *Post, Kitchell & Hodgson, 1998*). In some piscivorous fishes, young-of-the-year (YOY) intra-cohort individuals exhibited differential timing to the onset of piscivory, driving bimodal size frequency distributions and cannibalism in the age-0 group. In these cases, the larger, upper modal group (cannibals) increased their growth rate and decreased survival rate of the smaller, lower modal group (prey)

(*Buijse & Houthuijzen, 1992*; *Frankiewicz, Dabrowski & Zalewski, 1996*; *Post, 2003*; *Huss, Kooten & Persson, 2010*; *Heermann et al., 2014*). However, direct evidence of increased reproductive fitness with the transition to piscivory is absent, although *Post (2003)* suggested that an early transition to piscivory may result in higher lifetime fecundity since these fish have a larger relative size and potentially more spawning opportunities in a lifetime.

The clearhead icefish, *Protosalanx chinensis* (Abbott 1901), is a small, pelagic, euryhaline fish species of the family Salangidae that occurs in eastern Asia (*Xie & Xie, 1997*; *Saruwatari, Oohara & Kobayashi, 2002*). It is a short-lived semelparous fish, completing its lifecycle in about 1 year (*Tang et al., 2012*). Early-stage juveniles feed mainly on zooplankton, while larger individuals in young and adult-stage could switch to fish and shrimp and feed primarily on conspecifics (*Tang et al., 2013*). The size-frequency of the population becomes bimodal from month to month (*Zhu, 1985*). Considering the unique biological characteristics of *P. chinensis*, we present evidence of within-population size divergence, and the advantage of larger individuals over smaller individuals in survival and reproduction. These findings provide direct evidence of cannibalism, and its effects on reproductive fitness of both females and males.

## METHODS

### Study area

Lake Xingkai (Khanka) is a large shallow lake located between China and Russia, serving as a boundary. It is connected to the Ussuri River through the Sungacha River. The northern part of the lake (28% of the total area) is located in China and the southern part (72% of the total area) is located in Russia. The water level area in Khanka is variable and varies from 3,940 to 5,010 km². The long-term average water level area is 4,070 km² with an average depth of 4.5 m. Since 2010, there has been an increase in the water level in the lake. Lake Khanka/Xingkai is one of the most important international wetlands in the world. Specimens were sampled from the northwestern part of Khanka Lake near the locality of Dangbi in China(N45°15′49.12″–45°15′8.62″, E132°02′35.22″–132°01′7.73″, Fig. 1).

### Sampling procedure

Clearhead icefish (*P. chinensis*) were sampled monthly from March 2010 through March 2011 except for April 2010. Different sampling gear was used in winter and ice-free seasons to sample fish effectively. A cone trawler net (length, 3 m; diameter, 1 m; mesh size, 1 × 1 mm) was used for sampling fish in May, while a seine net (length, 200 m; height, 8 m; mesh size, 2 × 2 mm) was used for sampling from June through November 2010. For each sample, the seine net was drawn parallel to the shore where the water depth was less than 1.5 m, for about 50 m. Further, four gill nets (length, 30 m; height, 1 m; mesh size, 10 mm, 15 mm, 20 mm, and 23 mm between opposite knots) positioned randomly were deployed in gangs, tied end to end perpendicular to the shore for 24 h per month during the freeze-up period (March 2010 and December 2010 to March 2011). For the reproductive biology and cannibalism experiments, we used two extra gill nets (length, 100 m; height, 1

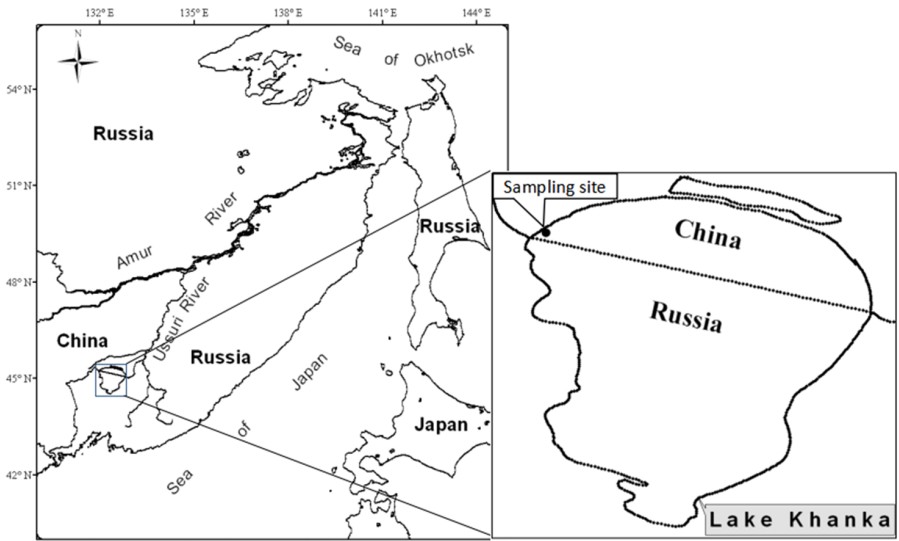

**Figure 1  Location of Lake Khanka (China/Russia) and the sampling site.**

m; mesh sizes, 20 and 23 mm between opposite knots) in the freeze-up period to obtain adequate samples.

## Life history traits analysis

The clearhead icefish died immediately after they were captured in nets. Over 100 specimens were caught in each monthly sample and were measured for standard length (SL; to the nearest 1 mm) and weight (WB; to the nearest 0.01 g). A row of scales typically appears at the base of the anal fin of maturing males and the anal fin also becomes larger and wave-shaped in mid-October, but no sexual dimorphism appears before that point, so the sex of each fish was determined by examining dissected gonads. Females have a larger ovary on the left of the ventral cavity and a smaller ovary on the right side, while males have a single testis on the right of the cavity. Sex ratios were determined, and their difference from parity was tested using a binomial test. Chi-square tests were performed to assess differences in sex ratios among months.

The gonads of females sampled in November, December, and January were used to determine the gonadosomatic index (GSI) and maturity stage, and more than 50 specimens were dissected for reproductive biology studies each month. In *P. chinensis*, the left ovary is usually larger than the right one. The middle portion of the left ovary was selected for histological examination andthe stages of maturity were classified into six phases according to *Sun (1985)*. Both ovaries were removed and weighed to the nearest 0.0001 g ($W_G$), and the GSI was calculated as follows:

$$GSI = 100 W_G / (W_B - W_G).$$

The number of oocytes both pre-vitellogenic and vitellogenic oocytes from dissected ovaries was counted in December and January. Two sub-samples of tissue weighing approximately 0.1 g were taken from the middle section of each ovary and preserved in 5%

formalin. The oocyte counts in these two sub-samples were determined using a dissecting microscope and were used to estimate fecundity if the coefficient of variation of N per unit ovary weight was below 5% for the two sub-samples. The total fecundity ($F_A$) and relative fecundity ($F_R$) were calculated in January as follows:

$$F_A = \sum Cn \, W_On / W_Sn \text{ and } F_R = 100 \, F_A / W_B,$$

where $Cn$ is the counted number of eggs in sub-sample n, $W_On$ is the corresponding ovary weight, $Wsn$ is the sub-sample weight and n is the number of subsamples.

To reveal the oocytes developed synchronously or asynchronously and the spawning pattern, the oocyte diameters frequencies were plotted. The Mann–Whitney U test was used to compare the developmental stages of gonads between the upper and lower modal groups in the same cohort within the maturing season. The difference in GSI between November, December, and January, as well as the difference in fecundity between December and January, was tested using analysis of covariance (ANCOVA) with WB as the covariate. Differences in GSI, fecundity, and relative fecundity between the upper and lower modal groups in January were examined using one-way analysis of variance (ANOVA). All statistical analyses were conducted using STATISTICA 12.0 (StatSoft, Tulsa, OK, USA). $P < 0.05$ was considered significant.

### Analysis of cannibalism

*P. chinensis* are translucent, allowing for gross examination of gut contents and visual identification of cannibalism in individuals. We chose to dissect cannibals with visible prey in their guts. Cannibals and prey were examined for SL (to the nearest one mm). The prey to cannibal size ratios in about 20 cases were calculated for each month from July through December.

## RESULTS

### Growth and intra-cohort size divergence

Two cohorts of *P. chinensis* were identified during the sampling period. Individuals collected in March 2010 belonged to the 2009 cohort and the individuals sampled after March 2010 belonged to the 2010 cohort. Samples collected in March 2010 ranged in SL from 81 to 225 mm and consisted of reproductive females and males from the 2009 cohort. *P. chinensis* juveniles belonging to the 2010 cohort were sampled from May 2010 onwards, and the SL distribution pattern was unimodal until early July, when a bimodal length distribution began to emerge and lasted through January 2011. From November to January, the reproductive population could be divided into the lower modal group (SL < 120 mm) and upper modal group (SL 120 mm) on the basis of size (Fig. 2). The population sampled in February and March did not exhibit a bimodal distribution because the fish died after spawning.

### Characteristics of cannibals and prey

Cannibalism was first observed in July. The SL of cannibals sampled during this month ranged from 65 to 91 mm, while those of prey ranged from 38 to 52 mm. Both cannibals and prey grew larger over time. The cannibals generally reached a SL of > 120 mm by

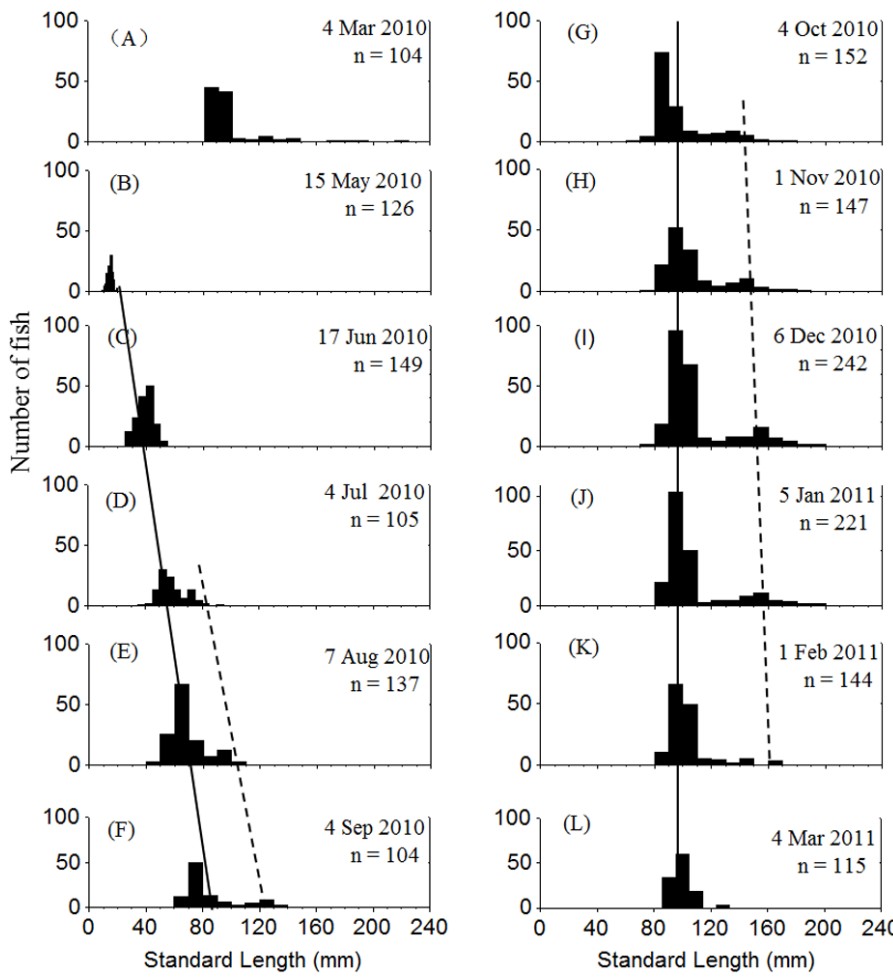

**Figure 2 Standard length frequency distribution of *P. chinesis* in Lake Khanka from March 2010 through March 2011.** The solid line connects lower modal groups and the dashed line connects upper modal groups. (A) Cohort born in 2009; (B–L) Cohort born in 2010. Jan, January; Feb, February; Mar, March; Jun, June; Jul, July; Aug, August; Sep, September; Oct, October; Nov, November; Dec, December.

November, and the prey were usually smaller than 120 mm (Table 1). Given that the reproductive population in November could be divided into two size groups by SL with 120 mm as the cutoff (Fig. 2), it is apparent that the upper modal group was comprised of cannibals and the lower modal group was comprised of the prey. The prey to cannibal size ratio ranged from 0.47 to 0.67 (Table 1).

## Examination of sex ratio

The sex of all sampled individuals was determined from early October 2010 to early March 2011. Monthly observations of sex ratios (male to female) did not differ from parity (1:1) from October through January ($P = 0.808, 0.410, 0.563, 0.621$, respectively) or vary significantly among months ($x^2 = 1.499, df = 3, P = 0.683$). However, the sex ratios in February and March were 13.4 and 18.1, respectively, due to natural mortality of females after spawning (Fig. 3). The remaining population mainly consisted of males in the lower

**Table 1 Differences in mean standard lengths (SL) ± standard deviation (SD) between _P. chinensis_ cannibals and prey.** Jul, July; Aug, August; Sep, September; Oct, October; Nov, November; Dec, December.

| Month | N | SL of cannibals (mm) | | SL of the prey (mm) | | SL of the prey / SL of cannibals | |
|-------|---|-------|-------------|-------|-------------|------------|-------------|
| | | Range | Mean (S.D.) | Range | Mean (S.D.) | Range | Mean (S.D.) |
| Jul | 24 | 65–91 | 76 ± 7.2 | 38–52 | 48 ± 6.2 | 0.54–0.65 | 0.63 ± 0.02 |
| Aug | 25 | 72–110 | 89 ± 13.5 | 46–59 | 51 ± 4.9 | 0.47–0.64 | 0.57 ± 0.05 |
| Sep | 29 | 108–146 | 128 ± 9.5 | 55–75 | 65 ± 5.5 | 0.48–0.54 | 0.51 ± 0.02 |
| Oct | 31 | 119–177 | 139 ± 16.1 | 72–102 | 83 ± 8.6 | 0.49–0.67 | 0.60 ± 0.05a |
| Nov | 27 | 122–189 | 147 ± 15.7 | 71–115 | 88 ± 10.2 | 0.51–0.65 | 0.60 ± 0.03 |
| Dec | 35 | 123–196 | 153 ± 16.7 | 82–117 | 93 ± 9.6 | 0.50–0.70 | 0.61 ± 0.04 |

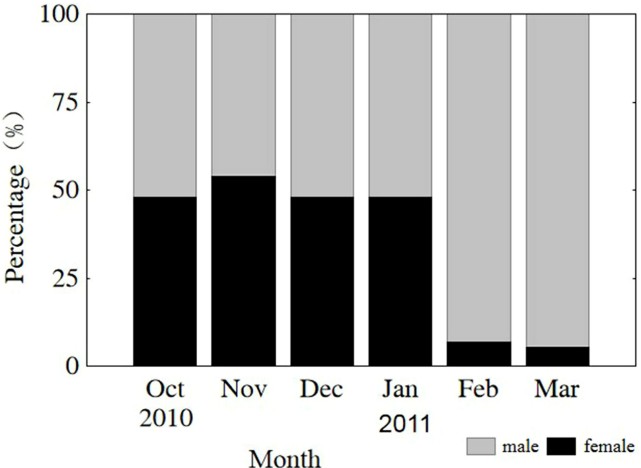

**Figure 3 Monthly sex ratio variation from 4 October 2010 through 4 March 2011.** Oct, October; Nov, November; Dec, December; Jan, January; Feb, February; Mar, March.

modal group (Fig. 2). From October through January, sex ratios did not differ from parity in the upper modal group ($P = 0.453, 0.450, 0.475, 0.527$, respectively) or the lower modal group ($P = 0.462, 0.409, 0.428, 0.635$, respectively).

## Gonad development and the differences between the two groups

Oocytes developed asynchronously in ovaries of _P. chinensis_; pre-vitellogenic (Phase IIIII) oocytes and vitellogenic (Phase IVV) oocytes existed in the same ovary. The gonads of female _P. chinensis_ did not develop to stage II until early October, after which they developed quickly to stage III in the first half of October. In early November, individuals in stages III and IV of gonadal development accounted for 61.5% and 38.5% of the female population, respectively. A few oocytes developed to stage V at the beginning of December, but all the oocytes in an ovary did not develop synchronously to the same stage. Most female gonads developed to stage IV or V and the fish could spawn in early January, but only a few post-spawning individuals were sampled in February and March (Fig. 4). The oocytes developed asynchronously, and oocytes of different developing phases may commonly

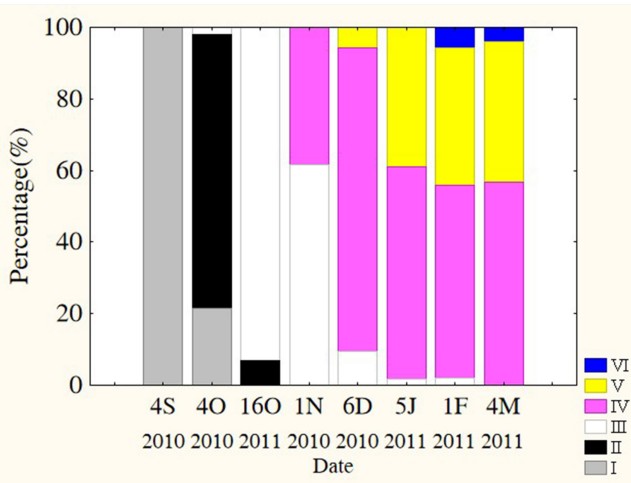

**Figure 4 Percentage of *P. chinensis* at different ovarian maturity stages from early October through early March.** Gonad maturity stage: *white*, I; *crossed*, II; *vertical lines*, III; *horizontal lines*, IV; *diagonal lines*, V; *black*, VI. S, September; O, October; N, November; D, December; J, January; F, February; M, March.

occur in the same ovary of maturing females (Fig. 5). Most female gonads developed to stage III or IV in early November, and females in the upper modal group seemed to mature faster than those in the lower modal group, although this difference was not significant ($Z = -1.117$, $P = 0.239$). In early December, females in maturity stage IV accounted for the largest proportion both in the upper (88.5%) and lower modal groups (81.5%), and the maturity stage distribution was not significantly different between the groups ($Z = -1.401$, $P = 0.161$). In early January, almost all of the female gonads developed to stage IV or V, and no significant differences were observed in the developmental rates between the groups ($Z = -1.934$, $P = 0.053$) (Fig. 6).

### Spawning pattern

Oocytes diameter distribution pattern was bimodal in December (Fig. 7A), which indicated that the oocytes developed asynchronously. However, the distribution pattern was unimodal in January (Fig. 7B). Results of the ANCOVA (BW as the covariate) also revealed no significant difference in fecundity between early December and early January ($F_{2,110} = 3.4$, $P = 0.067$). In other words, there was no down-regulation of fecundity during gonad development or indeterminate spawning. Therefore, *P. chinensis* is a semelparous fish although the oocytes developed asynchronously.

### Differences in reproductive investment between the groups

The GSI increased dramatically from November to January, ranging from 1.8 to 10.2% in November, 2.9 to 44.1% in December, and 7.6 to 76.4% in January (Fig. 8). Results of the ANCOVA (BW as the covariate) indicated that the differences in GSI among the three months were all significant ($F_{2,161} = 185.3$, $P < 0.001$). The GSI of fish in the upper modal group was significantly higher than that of fish in the lower modal group in November

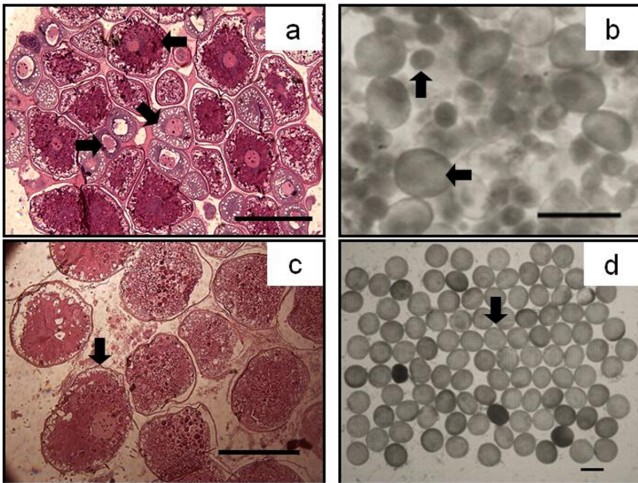

**Figure 5** **The asynchrony of oocytes development of *P. chinensis*.** Cross-section of ovary (A) and egg micrograph (B) of a fish in November (SL,145mm; BW,16.64g); cross-section of ovary (C) and egg micrograph (D) of a fish in January (SL,180 mm; BW, 26.15 g): (←) Phase II (Perinucleolar) oocyte; (↘) Phase III (Yolk vesicle) oocyte; (→) Phase IV (Yolk) oocyte; (↑) Pre-vitellogenic (Phase II&III) oocyte; (↓) Phase V (migratory nucleus) oocyte. Scale bar = 1 mm.

(ANOVA, $F_{1,50}$ = 56.3, $P < 0.001$), December (ANOVA, $F_{1,51}$ = 50.0, $P < 0.001$), and January (ANOVA, $F_{1,57}$ = 95.1, $P < 0.001$) (Fig. 8).

Samples from January were used to analyze intra-population divergence of fecundity. Fecundity ranged from 1,592 to 36,705 eggs, and relative fecundity ranged from 500 to 1,646 eggs per gram BW. The relative and absolute fecundity were significantly higher in females from the upper modal group than those in the lower modal group (ANOVA, $F_{1,57}$ = 121.5, $P < 0.001$ and ANOVA, $F_{1,57}$ = 21.94, $P < 0.001$, respectively) (Fig. 9).

## DISCUSSION

### Piscivorous, cannibalism and size divergence

Size variation among clearhead icefish populations has been previously reported (*Zhu, 1985*; *Wang & Jiang, 1992*; *Tang et al., 2003*). *Xie et al. (2001)* noted two peaks within the size frequency distribution of a reproductive population. However, this is the first report of within-cohort bimodal size distribution in *P. chinensis*. Bimodal size distribution is well documented in iteroparous fishes (*Huss, Kooten & Persson, 2010*; *Alejo-Plata, Díaz-Jaimes & Salgado-Ugarte, 2011*), and studies on Percidae species revealed diet shift contributing to the size divergences in YOY fish, and that the extreme growth of the upper modal group was induced by piscivory (cannibalism) (*Frankiewicz, Dabrowski & Zalewski, 1996*; *Post, 2003*; *Urbatzka et al., 2008*). The synchronous investigation on diet composition showed that a few bigger individuals of *P. chinensis* began to feed on fish and cannibalism also occured in early July, and then piscivory and cannibalism accompanied the size bimodal pattern forming (*Tang et al., 2013*). And the victims of cannibalism accounted for the majority of the diet of cannibals in December and January (*Tang et al., 2013*).

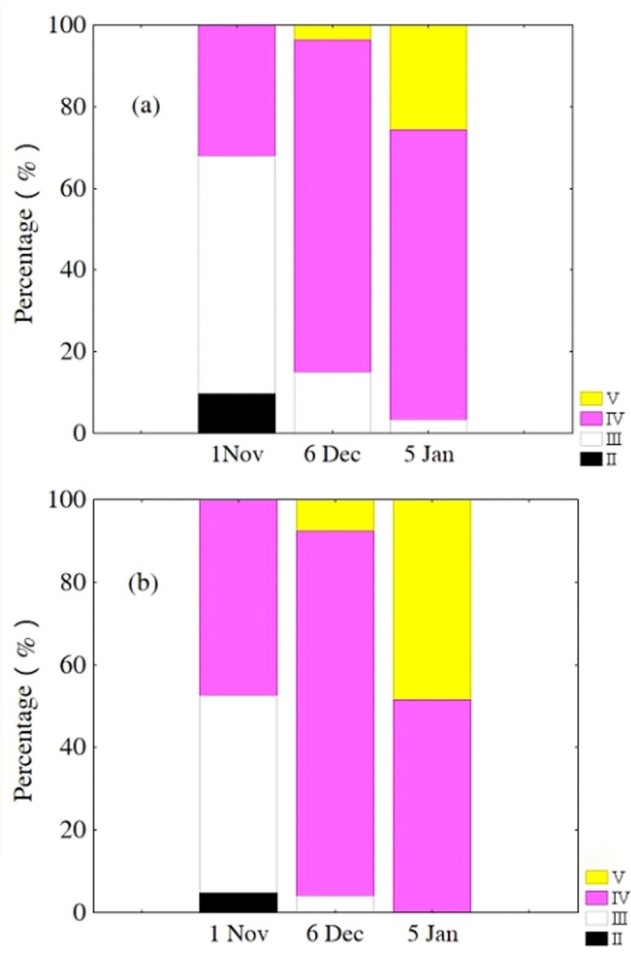

**Figure 6  Percentage of *P. chinensis* females with different ovarian maturity stages in early November, December 2010 and January 2011 from the lower modal group (A, SL < 120 mm) and upper modal group (B, SL > 120 mm).**

## Fitness of the cannibals

The absolute and relative fecundity reported in the present study for a population of *P. chinensis* in Lake Khanka shows wider range (1,592–36,705 eggs, and 500–1,646 eggs/gram-BW) than populations from other locations (3,295 to 34,520 eggs, and 340 to 1,290 eggs/gram-BW) (*Zhang et al., 1981*; *Xie et al., 2001*). While there is no published information illustrating the difference in reproductive investment between the two modal groups of clearhead icefish, this investigation demonstrates that the reproductive investment of females belonging to the upper modal group(cannibals) was significantly higher than that of females from the lower modal group(victims) in terms of GSI, absolute fecundity and relative fecundity. In common, larger fish has higher absolute fecundity and lower relative fecundity than the smaller conspecifics. *P. chinensis* cannibals have both higher absolute fecundity and relative fecundity than the victims, which means piscivory and cannibalism increased reproductive potential greatly. In addition, the cannibals grow
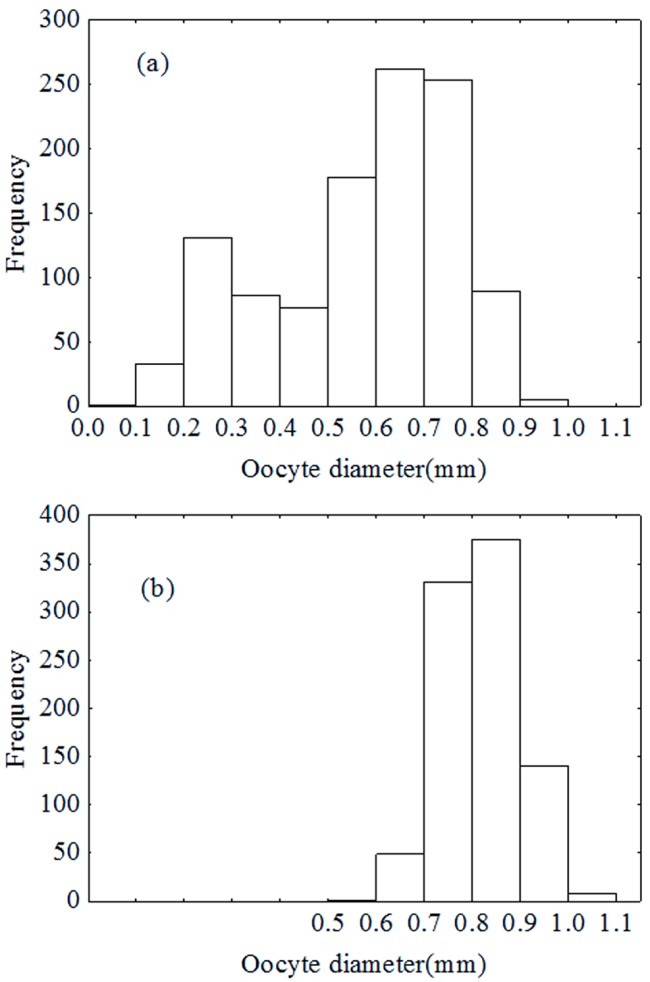

**Figure 7** **Oocytes development in December (A) and January (B).**

faster and become bigger than the potential victims, which will increase their survival rate undoubtedly.

## Gonad development and spawning pattern

Researchers have offered multiple theories to describe the reproductive strategy of *P. chinensis*. *Sun (1985)* reasoned that *P. chinensis* spawn twice during their lifetime upon examination of their gonad histology, but *Xu et al. (2000)* determined that this species spawns once before death. *Xie et al. (2001)* theorized that *P. chinensis* can spawn once or multiple times. Our research supports the theory that *P. chinensis* exhibits a semelparous reproductive strategy, although oocytes developed asynchronously in the same gonad. This is different from other semelparous species such as chum salmon *Oncoryhchus keta* (*Crozier et al., 2008*) and Sea lamprey *Petromyzon marinus* (*Mcbride et al., 2015*). The asynchronous development of oocytes should be attributed to the relatively rather high fecundity and the nutrition may not be enough to develop all the oocytes at the same time, which is a mechanism to produce as many eggs as possible in a spawning event in a life time.
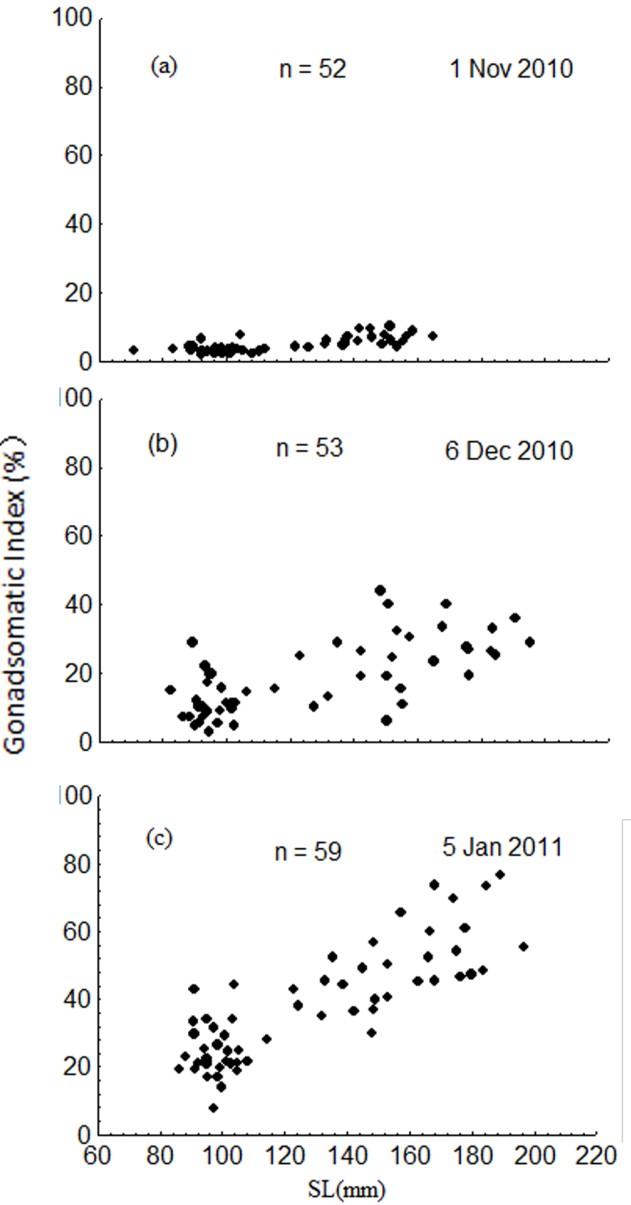

**Figure 8  Intra-population variation in gonadosomatic index (GSI) in November 2010 (A), December 2010 (B) and January 2011 (C).** SL = standard length.

## Sex ratio and mate selectivity

Sex ratios did not differ from parity before spawning (October through December) or during spawning (primarily early to mid-January) in both the upper modal and lower modal groups. This is similar to reports for this species from other locations (*Zhang et al., 1981*; *Sun & Zhou, 1989*; *Xu et al., 2000*; *Xie et al., 2001*). However, we found that the reproductive population was mainly composed of males in the lower modal group after January. We hypothesize that males in the lower modal group have a reduced opportunity

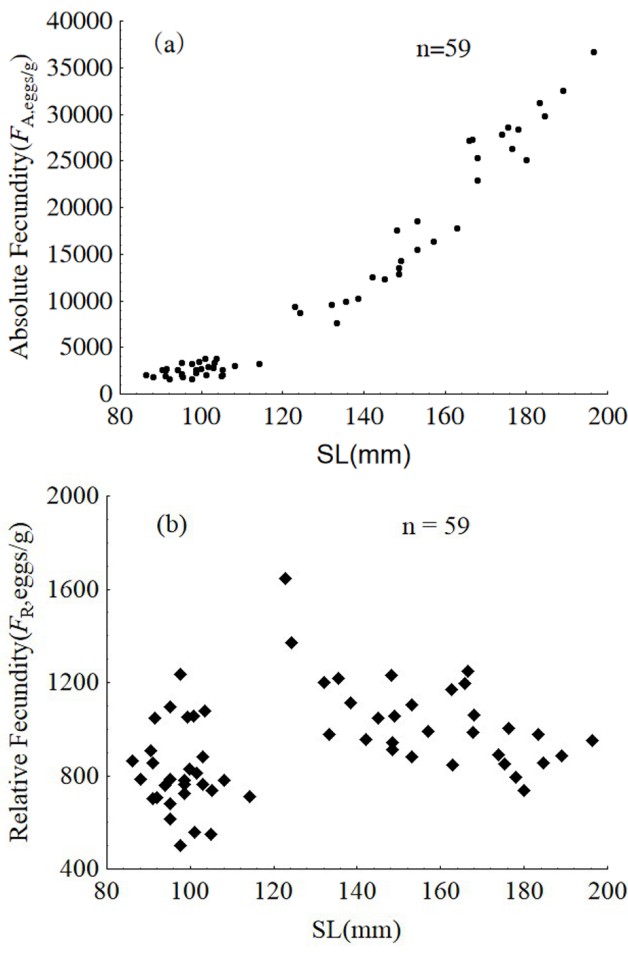

**Figure 9** **Absolute fecundity (A) and relative fecundity (B) of** *P. chinensis* **females sampled in January.** SL = standard length.

to participate in spawning. The intra-cohort sex selectivity of *P. chinensis* may influence evolution towards early birth and faster growth, although this theory requires further study.

## Life history strategy

The *P. chinensis* is a small fish that matures early with a short life span and relatively high fecundity. This makes it an exception from the three-point life-history continuum model proposed by *Winemiller & Rose (1992)*. Larger individuals (upper modal group) tend to exhibit an opportunistic life-history strategy, but characterized by relatively higher fecundity. An opportunistic colonization strategy facilitates range expansion of an invading species through early maturity in frequently changing or stochastic environments (*Fox, Vila-Gispert & Copp, 2007*; *Ribeiro, Collares-Pereira & Moyle, 2009*). Exceptional higher fecundity would undoubtedly contribute to rapid increase in population size and spread of the opportunistic fish, *P. chinensis*.

## CONCLUSION

Transition to feed on fish increased the investment of the female *P. chinensis* and may also increase the fitness of male in reproductive competing. This research provided a direct evidence that transition to be carnivorous is positively significant in evolution, although further study on increasing the fitness of male *P. chinensis* need to be conducted.

### Funding

This work was supported by the National Natural Science Foundation of China (grant number 31201993) and the Central-Level Non-profit Scientific Research Institutes Special Funds of China [grant number HSY201806M]. The funders had no role in study design, data collection and analysis, decision to publish, or preparation of the manuscript.

### Grant Disclosures

The following grant information was disclosed by the authors:
The National Natural Science Foundation of China (grant number 31201993) and:
31201993.
The Central-Level Non-profit Scientific Research Institutes Special Funds of China:
HSY201806M.

### Competing Interests

The authors declare there are no competing interests.

### Author Contributions

- Fujiang Tang conceived and designed the experiments, prepared figures and/or tables, and approved the final draft.
- Wei Liu conceived and designed the experiments, authored or reviewed drafts of the paper, and approved the final draft.
- Jilong Wang performed the experiments, authored or reviewed drafts of the paper, and approved the final draft.
- James Henne analyzed the data, prepared figures and/or tables, and approved the final draft.

### Data Availability

   The data are available in the Supplemental File.

### Supplemental Information

Supplemental information for this article can be found online at http://dx.doi.org/10.7717/peerj.9900#supplemental-information.

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
