# Peer review of "Life-history traits and intra-cohort divergence of clearhead icefish (Protosalanx chinensis), indicating cannibalism increased fitness"

_PeerJ, doi:10.7717/peerj.9900_

## Round 0.1 · original submission · Minor Revisions

Both referees liked and complimented your paper while making some good suggestions for improvements.

·

Basic reporting

.

Experimental design

.

Validity of the findings

.

Additional comments

Manuscript review

“Life-history traits and intra-cohort divergence of clearhead icefish(Protosalanx hyalocranius), indicating cannibalism increased fitness”

Fujiang Tang, Wei Liu, Jilong Wang, James Henne


When working with noodle fish, the presence of two size groups (cohorts) is noteworthy. Noodle fish is a monocyclic short-cycle species of fish that lives only one year and dies after spawning. The authors convincingly showed that the difference in growth rate is associated with a difference in the nutrition of fish. Fish that feed on individuals of their species have a higher growth rate than fish that feed on zooplankton.
The conclusions of the work correspond to the task. The results of the work are convincing and allow us to evaluate the reason for the appearance of two size cohorts in the noodle fish population of Khanka Lake.
This manuscript is written in fairly high quality and can be accepted for publication, but there are a number of comments and inaccuracies to it, which, in my opinion, should be eliminated.

Remarks.

1. Line 3 onwards. The name of this species used in the article is outdated. It is currently the junior synonym for the species Protosalanx chinensis (= Protosalanx hyalocranius). In this regard, it is necessary to bring the name of the species in line. Look site online Eschmeyer's Catalog of Fishes California Academy of Sciences (http://researcharchive.calacademy.org/research/ichthyology/catalog/fishcatmain.asp).

2. Line 73. The water level area in Khanka Lake is variable and varies from 3940 to 5010 km2. The long-term average water level is 4070 km2 with an average depth of 4,5 m. Since 2010, there has been an increase in the water level in the lake. This is not noted in the present work and references to the literature from where the data are taken are not given.

3. Line 74. Khanka Lake is not located in the upper reaches of the Ussuri River. It is located a little in the middle of the river and connects to the Ussuri River through the Sungacha River. The upper reaches of the Ussuri River is located on the slopes of Snezhnaya Mount, located about 200 km southeast of Khanka Lake.

4. Lines 77-78. Not exactly understandable and inaccurate phrase. Perhaps it should have been written that in 1996, an international Russian-Chinese reserve “Khanka Lake” was formed on Khanka Lake on the basis of the Khanka State Reserve (Russia) and the Xingkaihu Nature Reserve (China).

5. Line 79. To make it clear where the samples were taken, it was necessary to write: “in the northwestern part of Khanka Lake near the locality of Dangbi”.

6. Line 86. Not the “width” of the net, but the “height”.

7. Line 89. Not the “depth” of the branchial network, but the “height”.

8. Line 93. Not the “depth” of the branchial network, but the “height”.

9. Line 99-102. Not entirely accurate description. Males differ from females in the structure of the anal fin, the same picture is observed in capelin.

10. Line 171. This part of the work does not take into account the fact that Khanka Lake is a very large body of water. The sampling site on the Chinese side is located in the deep part of the lake. If we assume that the first fast-growing cannibal fish cohort spawns in January-March, then, according to Russian residents, spawning fish are caught in bulk in May in rivers in the south of the lake. This is most likely the second, slowly growing cohort. In this regard, the proportion of females in the north of the lake is reduced due to active migration to spawning sites in the southern part.
However, this work (of the Chinese-Russian specialists) needs to be carried out comprehensively throughout the lake, as This issue has not been closed. The authors reflected in the manuscript what they observed in their part of the lake.


I hope that the work of the authors in this direction will be continued and reflected in new articles.




Head of the Laboratory
of Biological Resources
of Continental Waterbody
and Fish of Estuarine Systems
Pacific Branch “VNIRO” (“TINRO”), PhD Evgeny Barabanshchikov

·

Basic reporting

Overall, this reads well. One exception is the ambiguous terminology applied for maturity classification, which on l. 108-109, it is only stated that “Stages of maturity were classified into six phases according to Sun (1985).” Sun (1985) does not come up in an internet search, so it is not accessible, and thereby it is not possible to examine whether it conforms with newer practices (e.g., Brown-Peterson et al., 2011; DOI: 10.1080/19425120.2011.555724 or McBride et al. 2015; http://dx.doi.org/10.1111/faf.12043). Worse, in the manuscript, the maturity stages in Figures 4 and 7 are numerical, without any label or context. I would recommend that labels are associated with the Roman numeral classes of maturity and some attempt is made to show that this scheme still makes sense relative to newer references. The same can be said about phases of the oocytes, although these at least have labels associated with them (e.g., l 177-8)

Experimental design

This paper employs a complementary set of relatively simple methods. Nice job. I thought carefully about the potential for different selectivities of the multiple gears. This could perhaps be addressed more directly by the authors, but I don’t see that any problems would emerge from a more sophisticated analysis. The sampling frequency is good.

Validity of the findings

I am not so sure about your conclusion that oocyte development in asynchronous. That is unlikely for a semelparous fish (see for example McBride et al. 2015; http://dx.doi.org/10.1111/faf.12043. More likely, in Fig. 5, there are a mix of closely related oocyte phases early in development (5a) but these become more of a single, advanced phase closer to the spawning event. I encourage you to look closer at your data, examining a few more samples as appropriate, to reach your conclusion about the synchrony of oocyte development in ice fish.

I would be careful about confusing reproductive investment (e.g., GSI) or reproductive potential (absolute fecundity) with fitness (at least defined as producing offspring that go on to reproduce themselves). Do you know anything about the breeding system? Here is one possibility: What if the small males can ‘sneak’ in on a broadcast spawning event and successfully fertilize eggs? There are many other possibilities. I like the system you have revealed to me here, but to address the fitness issue more work is needed, like a parentage assessment.

You mention that these results suggest ‘evolution towards early birth and faster growth’ (l. 265). Perhaps, but if so, is warming occurring over time in your region and how may this accelerate or dampen this possibility. Also, consider that the constant sex ratios early on, preclude environmental sex determination from causing your patterns of size and sex (e.g., Conover 1984; https://www.jstor.org/stable/2461097). You might want to mention that.

Additional comments

l. 58, 64, 81, and throughout: put the species name in Italics

l. 63 define ‘hierarchical growth’ and put it in context

l. 114 What stage(s) were these oocytes that were counted? Are you counting only a specific stage (e.g., vitellogenic) or multiple stages (e.g., the leading size cohort)?

149-150 there is something wrong with this text “the LS distribution pattern was unimodal (range, 10-20 mm) until early July,” What is LS (SL?) and is that range correct?

l. 170 you refer to only natural mortality here. Is that true: that is, there is no direct icefish fishery or bycatch fishery from some fishing gear in this lake?

Fig. 3. What are the arrows in a-d?

Fig. 7. Should ‘b’ be the upper modal group (not lower)?

---

## Round 0.2 · Minor Revisions

Sorry I cannot follow the rebuttal. You attribute comments to me that I did not make and ask me questions about them I cannot answer. This documents needs to clearly distinguish the "Referees comment" from your "Response". It is also confusing formatted (single spacing is ok. Please tidy it up and resubmit.

---

## Round 0.3 · Minor Revisions

In future I suggest you improve the formatting of the 'rebuttal'. Clearly separating the referees' comments and your responses (e.g., skip a line, or use different colour text). There is no need for size 14 font and double spacing, font 11 or 12 and single spacing is ok. It was also confusing with line numbers and response numbers and starting sentences with "l." (say 'Line ..").

Please consider these further minor corrections and improvements to figures:
There was a question about citing data sources - yes you should.
You say you cannot change something in the online system regarding figure. I do not understand this. the arrows do seem to be on the pdf of the MS. So are the legends with each figure. Please contact the PeerJ staff during final submission for any difficulties in this regard. In figures there should also be a character space before around parentheses; (e.g. Fig 6) there should be a space between "diameter(..." and legend fig 7 "overianmaturity" is all one word in error (see legend fig 6, 9 for similar mistake).
Table 1 legend still uses the old name of the species. I doubt the colours in fig 4 and 6 will be easily visible to people with colour blindness. the background colour is best removed to keep figures simple and clear, and simple hatching and stippling used to distinguish parts of the maturity stages.

---

## Round 0.4 · Minor Revisions

Thank you for the revised paper. Please see the suggestions of one referee regarding how synchronous is worded and some possible typographical errors. A referee also provided a paper that may interest you. It is optional whether you cite it or not.

·

Basic reporting

.

Experimental design

.

Validity of the findings

.

Additional comments

Everything is fine, the authors corrected the comments. The article is good, I hope that the authors will continue to study the life cycle of this species of fish.

·

Basic reporting

see below

Experimental design

see below

Validity of the findings

see below

Additional comments

This is an improved draft but I still have some concerns, outlined below, that the authors should address before going to print.


l. 66-67 This sentence (Size-frequency …as growing (Zhu 1985)) is better but could be improved. How about: “The size-frequency of the population becomes bimodal from Month to Month” or “The size-frequency of the population become bimodal during Name the season, when the fish are approximately X-X months old” citing Zhu 1985.
l. 193-203 I still believe you are describing that the oocytes develop synchronously. I understand that they are not advancing together—all at once—from one stage to the next. That would be unexpected for two reasons: 1) the stages are a rank variable system that describes what is fundamentally a continuous process, and 2) there are minor deviations between germ cells in their rate of development. Instead, what you are describing here is likely a continuous, normally-distributed mode of oocytes that advance together (synchronously). For example, in Figure 6, the proportion of pre-vitellogenic cells (Pp) is declining at a linear rate because the tail of the normal distribution is becoming smaller and smaller in a gradual manner. In terms of testing the hypothesis of asyn- versus synchrony, it is not necessary for the decline in Pp to occur all at once for it to be synchronous. This difficulty in distinguishing asyn- from synchronous development is the authors do not plot the oocyte diameter frequency. Again, look at McBride et al. (2015; attached) in Figure 3 for the expected differences in oocyte diameter distributions between these two types of oocyte development.
l. 207 Is P really equal to 0? More likely it is < 0.001
l. 211 I am not convinced that the decline in pre-vitellogenic oocytes means that the fish will spawn in a single spawning event. It is still possible that vitellogenic ooyctes will mature and ovulate in multiple batches. Instead, if you are really seeing a depletion of pre-vitellogenic oocytes, that likely means this fish is semelparous because it does not maintain a reservoir of pre-vitellogenic oocytes for future years of spawning. Again, see attached, McBride et al. (2015) Figure 3 for an example.

-Richard McBride

---

## Round 0.5 · Minor Revisions

Dear author,

Please note that your Responses to referees reviews are not aimed at the referees. Rather they are to the Editor. We cannot make changes to the MS for you. Unfortunately, I cannot understand your responses. It does not seem you have understood the referees' comments and amended the text to clarify it. There were only 4 items to address.

1. If my memory is correct your previously used the word 'herarchical' and it was ambiguous (I cannot understand what it means in this context) and you have not followed any of the referees suggestions.
2. So how have you addressed this in the MS?
3. Regarding P= or P< please use P<.
4. So how have you clarified or addressed this in the MS?

yours sincerely,
Mark Costello

---

## Round 0.6 · Minor Revisions

I appreciate that English is not your first language so I rechecked the MS with your responses and the referee's comments. More explicit comments with regard to what you have or have not changed in response to the referees' comment would be really helpful to the editor. It seems you disagree with the referee and believe the fish spawn synchronously in one event because the oocytes are lost within two months. It is not clear to me how the speed of egg loss tells us whether the spawning was in one or multiple events (e.g. every few days). Can you please clarify and be clear in why and what you are doing?

---

## Round 0.7 · Minor Revisions

I ask for a third time what is your action regarding the referees comments. It would be really helpful if you clearly stated what you were going to do or not do. See your responses below – you do not say if you have changed the MS in response to the referees suggestions or not. Are you agreeing or disagreeing with the editor?

l. 193-203 I still believe you are describing that the oocytes develop synchronously. I understand that they are not advancing together—all at once—from one stage to the next. That would be unexpected for two reasons: 1) the stages are a rank variable system that describes what is fundamentally a continuous process, and 2) there are minor deviations between germ cells in their rate of development. Instead, what you are describing here is likely a continuous, normally-distributed mode of oocytes that advance together (synchronously). For example, in Figure 6, the proportion of pre-vitellogenic cells (Pp) is declining at a linear rate because the tail of the normal distribution is becoming smaller and smaller in a gradual manner. In terms of testing the hypothesis of asyn- versus synchrony, it is not necessary for the decline in Pp to occur all at once for it to be synchronous. This difficulty in distinguishing asyn- from synchronous development is the authors do not plot the oocyte diameter frequency. Again, look at McBride et al. (2015; attached) in Figure 3 for the expected differences in oocyte diameter distributions between these two types of oocyte development.
Response: I have checked the data in December and January and plotted the oocyte diameter frequency. The oocyte diameter distribution frequency in December showed the same as Example 3 in Figure 3 (McBride et al. 2015), but that in January showed the same as Example 1 in Figure 3 (McBride et al. 2015). See them below:

Therefore, the oocytes development asynchronously in the same ovary due to the rather high relative fecundity without enough nutrition at the same time, but they won’t be spawned untill all of them were well developed when a spawing event happened. The data showed there wasn’t a decline of oocytes number between January and December which was mentioned in the MS. The oocyte development and spawning pattern of Protosalanx chinensis should be an exception to the three examples of the paper (McBride et al. ,2015) and a new insight into the reproductive biology of fish. therefore, this paper is of high value, which will also stop the arguments existing on this issue of P. chinensis.

l. 211 I am not convinced that the decline in pre-vitellogenic oocytes means that the fish will spawn in a single spawning event. It is still possible that vitellogenic ooyctes will mature and ovulate in multiple batches. Instead, if you are really seeing a depletion of pre-vitellogenic oocytes, that likely means this fish is semelparous because it does not maintain a reservoir of pre-vitellogenic oocytes for future years of spawning. Again, see attached, McBride et al. (2015) Figure 3 for an example.
Response: In Figure 6, we can see there is a depletion of pre-vitellogenic oocytes in most of the individuals which showed a maturing trend and meant they will spawn in a single spawning event. The decline in pre-vitellogenic oocytes means that the immature oocytes are developing in this period, which means they will mature and be spawned shortly but not waiting to the next year or season. This fish is semelparous

---

## Round 0.8 · Minor Revisions

I will try another way to get clarity by adding comments to your response on attached file. It is still not clear to me whether you are agreeing or disagreeing with the referee, and if so exactly what changes if any were made to the MS.

---

## Round 0.9 · accepted · Accept

Thank you for clarifying the changes, and the changes that were not made and no longer necessary.